# Application of Three-Dimensional Culture Method in the Cardiac Conduction System Research

**DOI:** 10.3390/mps5030050

**Published:** 2022-06-14

**Authors:** Abhishek Mishra, Kishore B. S. Pasumarthi

**Affiliations:** Department of Pharmacology, Dalhousie University, Halifax, NS B3H 4R2, Canada; mishraak@dal.ca

**Keywords:** three-dimensional culture, embryonic cardiac cells, ventricular conduction system

## Abstract

Congenital heart defects (CHD) are the most common type of birth defects. Several human case studies and genetically altered animal models have identified abnormalities in the development of ventricular conduction system (VCS) in the heart. While cell-based therapies hold promise for treating CHDs, translational efforts are limited by the lack of suitable in vitro models for feasibility and safety studies. A better understanding of cell differentiation pathways can lead to development of cell-based therapies for individuals living with CHD/VCS disorders. Here, we describe a new and reproducible 3-D cell culture method for studying cardiac cell lineage differentiation in vitro. We used primary ventricular cells isolated from embryonic day 11.5 (E11.5) mouse embryos, which can differentiate into multiple cardiac cell types including VCS cells. We compared 3-D cultures with three types of basement membrane extracts (BME) for their abilities to support E11.5 ventricular cell differentiation. In addition, the effects of atrial natriuretic peptide (ANP) and an inhibitor for its high affinity receptor were tested on cell differentiation in 3-D cultures. Following the cell culture, protocols for immunofluorescence imaging, cell extraction and protein isolation from the 3-D culture matrix and in-cell western methods are described. Further, these approaches can be used to study the effects of various ligands and genetic interventions on VCS cell development. We propose that these methodologies may also be extended for differentiation studies using other sources of stem cells such as induced pluripotent stem cells.

## 1. Introduction

A diverse array of genetic and structure-function changes has been implicated in the etiology of congenital heart defects (CHDs) and cardiac conduction system disorders [1,2,3]. While the outlook for patients living with these disorders is much better with improvements in surgical and medical management, these patients tend to develop arrhythmias and heart failure [4]. Advances in cardiac development and stem cell fields hold promise for the treatment of CHDs and conduction system abnormalities [5].

In mouse embryos, ventricular conduction system (VCS) development begins after embryonic day 11.5 (E11.5) and a fully functional VCS appears after E17.5 [6,7]. The gap junction protein, connexin 40 (Cx40) expression is uniform in the trabecular myocardium at E11 stage and it becomes restricted to the VCS and atria by E16 through neonatal stages [8,9]. There is evidence that factors secreted from the coronary endothelium, endocardium, and trabecular myocardium (e.g., endothelin-1, neuregulin-1 and atrial natriuretic peptide) provide instructive cues for the VCS cell fate [10,11,12,13]. Although two-dimensional (2-D) cultures of embryonic ventricular cells and embryonic stem cells have been used to study cell proliferation and differentiation patterns [14,15,16,17,18], mechanisms regulating VCS cell formation are not very well characterized. Notably, Purkinje fibers form a complex network of peripheral VCS components from base to apex in the subendocardial regions of the mammalian heart [19]. Development of novel cell culture models which permit formation of three-dimensional (3-D) networks can facilitate a better understanding of mechanisms regulating VCS development and function.

When compared to conventional 2-D cell cultures, the 3-D cultures capture the complexity and architecture of cultured cells in vitro and mimic more closely to the native tissue milieu. This could help with predicting appropriate responses to the biochemical, mechanical, and electrical stimuli, and lead to physiologically relevant gene expression profiles [20,21,22]. Earlier studies have reported excellent 3-D culture protocols for growing cardiac cells derived from pluripotent stem cells and primary cardiac tissue [22,23,24,25]. The culture of chick embryonic ventricular cells in a 3-D collagen matrix allowed measurement of the isometric force of contraction [21]. The formation of cardiac fibers in a 3-D model using the Matrigel matrix offers another approach for structure and function studies [23].

The commercial forms of Basement Membrane Extracts (BMEs; e.g., Matrigel, Cultrex and Geltrex) are soluble forms of basement membrane extracts purified from the murine Engelbreth–Holm–Swarm (EHS) tumors [26]. These extracts are hydrogels containing multiple extracellular matrix components (e.g., laminin, collagen IV, enactin, heparin sulfate proteoglycans) that can polymerize at 37 °C to form a basement membrane. The BMEs with different levels of growth factors can be used for cell attachment, cell differentiation, angiogenesis, and cell migration/invasion assays. In comparison to BMEs, which are well suited for both 2D and 3D cultures, extracellular matrix (ECM) preparations provide defined and purified substrates (e.g., collagen I, fibronectin) for cell attachment in 2D cell cultures. The ECM proteins interact with cell surface integrins and these interactions are critical for various cellular processes including cell spreading, proliferation and differentiation [27].

An extensive survey of literature revealed no detailed 3-D culture protocols for studying cardiac developmental events and mechanisms regulating proliferation and differentiation of VCS cell lineage. In this report, we have modified a 3-D culture method originally developed for culturing normal and malignant tumor cells using Matrigel [20,28] and provided a detailed protocol for generation of VCS cells. Specifically, the protocol described here is focused on culturing of E11.5 mouse embryonic ventricular cells using different commercial BME matrices to study VCS cell formation. The E11.5 ventricular cells were shown to harbor a large number of bi-potential cardiac progenitor cells that can differentiate into working cardiomyocytes and VCS cells in 2-D cultures [11,15]. These cells were also shown to have higher cell proliferation, migration and engraftment efficiencies compared to ventricular cells from later developmental stages [29,30,31]. We standardized the number of embryonic cells and the culture duration required for generation of optimal contractile 3-D cultures. Subsequently, methods for VCS cell differentiation analysis in 3-D cultures were described.

## 2. Experimental Design

### 2.1. Materials and Reagents

#### 2.1.1. Culture Vessels

8-Well chamber slides (Nunc™ Lab-Tek™ II CC2™ Chamber Slide System, Cat# 154941).CELLSTAR μClear 96-well flat bottom microplates for cell culture (Greiner Bio-One 655090, Fisher Scientific, Ottawa, ON, Canada, Cat#07000166).

#### 2.1.2. Growth Media, Supplements and Drugs

Dulbecco’s Modified Eagles Medium (DMEM; Wisent, Cat#319-005-CL)10%. Fetal Bovine Serum (FBS; MilliporeSigma, Oakville, ON, Canada, Cat#F1051).1× Antibiotic/Antimycotic (1× Ab/Am; Gibco, Thermo Fisher, Mississauga, ON, Canada, Cat#15240-062)Atrial natriuretic peptide (ANP; Bachem, Torrance, CA, USA, Cat#4030380)A71915 (A7, Natriuretic peptide receptor A inhibitor; Bachem, Cat#4030385)

#### 2.1.3. Cell Isolation Reagents and Basement Membrane Extracts

0.2%. *v*/*v* type I Collagenase (Worthington Biochemical Corp., Lakewood, NJ, USA, Cat#LS004196)Cultrex PathClear Basement Membrane Extract (BME), (R&D System, Minneapolis, MN, USA, Cat#3432-005-01), stock concentration: 12–18 mg/mL.Cultrex PathClear Reduced Growth Factor BME, (R&D System, Cat#3433-001-01), stock concentration: 12–18 mg/mL.Gibco™ Geltrex™ LDEV-Free Reduced Growth Factor Basement Membrane Matrix, (Fisher scientific, Cat#A1413201), stock concentration: 12–18 mg/mL.Fibronectin (MilliporeSigma, Cat#F1141), stock concentration 1 mg/mL.

#### 2.1.4. Cell Culture Fixing

*4%. Paraformaldehyde (MilliporeSigma, Cat# P-6148)*.

#### 2.1.5. Immunostaining, Western Blotting and In-Cell Western Reagents

Connexin 40 (Cx40)/Gap junction alpha-5 Protein (CXA-5) antibodies- Rabbit Anti-mouse Cx40 IgG (Alpha Diagnostic International, San Antonio, TX, USA, Cat# Cx40-A); Western blotting—used at 1:660 dilution (1.5 µg/mL). Immunocytochemistry—used at 1:150 dilution (6.67 µg/mL).Sarcomeric myosin antibody. (Developmental Studies Hybridoma Bank, Iowa, Cat#MF20). Immunocytochemistry—1:150 dilution (0.15 µg/mL)Pan-Cadherin antibody (Thermo Fisher Scientific, Cat#71-7100), Western Blotting—1:200 dilution (1.25 µg/mL)Anti-rabbit IgG (H+L) secondary antibody, Alexa Fluor 488 (Thermo Fisher, Cat#A21206), 1:150 dilution (13.3 μg/mL).Anti-rabbit IgG (H+L) DyLight 800 PEG conjugate (New England Biolabs, Whitby, ON, Canada, Cat#515P), 1:150 dilution (6.67 μg/mL).Anti-mouse IgG (H+L) DyLight 680 conjugate (New England Bioloabs, Cat#5470P), 1:150 dilution (6.67 μg/mL).Hoechst Dye 33,258 (MilliporeSigma, Cat#14530), 1 mg/mL stock, 1 in 20,000 dilution.Immunofluorescence (IF) blocking solution [IF buffer with 1% Goat anti-mouse F(ab’)2 IgG (Thermo Fisher, Cat#31166) and 10% goat serum (Thermo Fisher, Cat#16210064)Immunofluorescence (IF) Buffer (0.2% Triton X-100, 0.1% Bovine Serum Albumin and 0.05% Tween-20 in PBS with 7.7 mM sodium azide; store as aliquots at −20 °C for long term usage)PBS-glycine (100 mM Glycine in PBS)

#### 2.1.6. Cell Extraction and Protein Extraction Reagents

Cell wash Buffe, (Thermo Fisher scientific, Cat#SD251321)Phosphate Buffer Saline- (PBS; 0.138 M NaCl, 0.0027 M KCl, pH 7.4)PBS-EDTA (5 mM EDTA, 1 mM Sodium Orthovanadate and 1.5 mM Sodium Fluoride in PBS)Lysis Buffer [1% NP40, 5 mM EDTA, 150 mM sodium chloride and 50 mM Tris-HCL, pH 8.0 in PBS supplemented with phosphatase and protease inhibitor cocktail (Thermo Fisher, Cat#1861281) at 1:100 dilution and Aprotinin (MilliporeSigma, Cat#A6279) at 1:1000 dilution].

### 2.2. Equipment

Cell Culture Hood, class II Type A2 Biological Safety CabinetCell Culture Incubator (5% CO_2_, 37 °C Temperature, Model#3110, ThermoForma)Centrifuge (Eppendorf, Model#5415D)HemocytometerInverted light microscope (Leica DMIL) with a camera (Leica DFC290HD) installedSonicator (Model #100, Fisher Scientific)Zeiss LSM710-Laser Scanning Confocal MicroscopeLI-COR imaging system (Model#Odyssey CLx)Polarstar Omega Fluorescence Plate Reader (BMG Labtech)

## 3. Procedure

### 3.1. Storage and Handling of BME Matrix

#### 3.1.1. Aliquoting and Freezing of BME Matrix


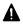
** CRITICAL
STEP:** Frequent thawing of the matrix can lead to loss of polymerization efficiency. To avoid multiple freeze thaw cycles, the stock solution of BME matrix can be aliquoted into multiple tubes and stored at −80 °C. This procedure must be performed on the ice platform or any chilled surface. All tubes and pipette tips must be pre-chilled and placed on ice during aliquot preparation.


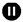
** PAUSE STEP**: All the aliquoted tubes must be stored at −80 °C immediately for future use.

#### 3.1.2. Thawing of BME Matrix


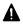
** CRITICAL STEP:** BME matrix is thawed overnight in the 4 °C environment prior to starting a 3-D culture experiment. Because refrigeration temperatures may vary, the BME aliquots should be kept on ice in the refrigerator during the thawing process.


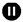
** PAUSE STEP**: Thawed BME polymerizes rapidly at temperatures above 15 °C. It is important to keep it on ice to prevent its solidification prior to coating culture vessels. Pre-chilled pipette tips and tubes must be used to prevent untimely gelling of BME during the coating procedure.

### 3.2. Embryonic Ventricular Cell Isolation

Embryos (E11.5) from the timed-pregnant CD1 mice (Charles River Laboratories, Senneville, Quebec, QC, Canada) are taken out from the uterus by removing the placenta and kept in dishes containing PBS with 1× Ab/Am. The whole heart is removed from the embryo, and then the atria and outflow tracts are dissected out.Both ventricles from each embryonic heart are placed in 0.2% *v*/*v* type I Collagenase in 1× PBS (~10 ventricles in 1 mL) and incubated for 45 min at 37 °C on a rocking platform and the ventricular tissue is digested.


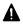
** CRITICAL STEP:** Collagenase digestion of E11.5 ventricles should not exceed 45 min for optimal cell viability and recovery.


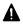
** CRITICAL STEP:** After 45 min of incubation, the tissue is triturated using a pipette tip for mechanical dissociation of the remaining tissue. Subsequently, dissociated tissue fragments are allowed to settle for 5 min for gravity separation of the fibrous tissue debris.

The supernatant is recovered, leaving debris in the tube, and the aqueous fraction containing cells is centrifuged at 1600 rcf (Eppendorf centrifuge, model#5415D) for 4 min at 4 °C. The supernatant is discarded, and the cell pellet is neutralized by two washes of culture media (DMEM containing 10% FBS and 1× Ab/Am).Finally, cells are resuspended using 10% FBS-DMEM, and the cell number is determined using a hemocytometer. The cell suspension is resuspended in media to achieve desired cell density for 3-D cultures.


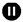
** PAUSE STEP**: Dispersed cells can be stored on ice for 30 min to 2 h before plating on BME matrix. However, it is recommended to plate cells immediately after isolation.

#### 3.2.1. Embryonic Ventricular 3-D Cell Culture over the Matrix Bed

Coat pre-chilled culture surface of 8-chambered slide with a thick layer of BME matrix in a sterile environment. The volume of the matrix needs to be optimized, but 60 µL/chamber is a good starting point. Pipette the matrix in the center of the chamber and spread uniformly over the surface using a pre-chilled pipette tip. Carefully spread the matrix in all the corners of the chambered slide.

**OPTIONAL STEP: 8**-Chambered culture slides and pipette tips can be kept in −20 °C for pre-chilling few hours before the experiment.


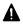
** CRITICAL STEP:** BME matrix polymerizes and solidifies rapidly at room temperature. Working over ice platform or any chilled platform in the culture hood delays the polymerization process and makes the uniform spread of the matrix in chambers effortless. Uniform spreading of the matrix is a critical factor in this method. Uneven matrix bed can lead to cell accumulation in areas with a thin layer of matrix.

**Note:** Matrix provides an optimal milieu for fibers to grow in any direction. The thickness of the matrix bed is a regulatory factor in the formation of cardiac fibers. A very thin matrix bed does not support fiber formation; it instead leads to formation of cell clusters without connections. A thick matrix bed promotes a good cardiac fiber structure formation. However, a thick matrix bed presents some practical difficulties such as mounting a coverslip after the staining procedure and poor visualization under the higher magnification using a confocal microscope. The thickness of the matrix bed can be decided based on the procedures to be done after culture. For protein extraction and RNA isolation procedures, the thickness should be more (3–5 mm), and for immunostaining purposes, the thickness should be optimally less (1–2 mm).

Let the coated surface dry for 15–30 min at 37 °C in the incubator but avoid over-drying to prevent dehydration.In the meantime, resuspend 125,000–250,000 cells in 100 µL of culture media (DMEM-10%FBS) per chamber. This volume is half of the complete media volume to be put in one chamber.After incubation of the pre-coated culture surface, pipette the mixture of embryonic ventricular cells and media over the matrix bed and incubate for 30 min in a cell culture incubator at 37 °C and 5% CO_2_.Chill the remaining media on ice and mix 10% volume of the matrix (100 µL media + 10 µL matrix/chamber). Gently mix this mixture homogeneously and pipette over the incubated culture along the wall of the chamber.Incubate the culture at 37 °C and 5% CO_2_ in the incubator. Depending on the experiment, treat the culture with desired drugs either during seeding of cells or a day after when the 3-D structure begins to form.Monitor the growth of culture under an inverted phase-contrast microscope and observe for the formation of three-dimensional structures (Figure 1C–G). Cell clusters formation starts in 2–3 days and connecting extensions between the contractile spheroid cell bodies can be observed in 7–10 days (Figure 1A–G). A healthy culture can be maintained for 13–15 days.

#### 3.2.2. Alternative Method of 3-D Culture, Cells Embedded in Matrix

Similar to the cells on top of the matrix bed protocol, coat the pre-chilled culture surface of an 8-Chambered slide with a thin layer of BME matrix. Evenly spread 30 µL of the matrix per chamber. Allow the coat to dry in the incubator at 37 °C for 15–30 min. Care should be taken as described in the earlier method.In the meantime, resuspend the isolated embryonic ventricular cells into an appropriate volume of the matrix. Optimize 150 µL matrix for 125,000–250,000 cells per chamber. The volume of the matrix can be standardized depending on the cell number used in the experiment. A higher number of cells need to be diluted with more volume of the matrix.


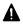
** CRITICAL STEP:** Resuspension of cells in the matrix is a critical step in cell embedded 3-D culture method. Resuspension needs to be done over the ice or any chilled platform to delay matrix polymerization. This step requires the mixing of cells in the matrix by pipetting, but it needs extra caution to prevent air bubble formation. These bubbles/vacuoles in the polymerized matrix may inhibit the formation of connecting fibers between the spheroid cell bodies produced during culture.

After polymerization of the pre-coated culture surface, pipette the mixture of cells and matrix over the pre-coated surface. Gently spread the mixture all over the surface in uniform thickness.Incubate for 10–15 min at 37 °C and 5% CO_2_ in the incubator to allow polymerization. Add the 200 µL culture media (DMEM-10%FBS) over the matrix in each chamber.If required, treat the culture with pharmacological agents as per the intended objective and then incubate at 37 °C and 5% CO_2_ in the incubator.Monitor the health of the culture under the inverted phase-contrast microscope and note the development of three-dimensional spheroid cell bodies (Figure 1A–G).

### 3.3. Feeding the Culture and Drug Testing

The 3-D cultures are fed at a regular interval of 36–48 h. Details of drugs/agents tested, dosage and time points for media and drug additions have been indicated in a timeline chart (Table 1). Depending on the half-life of drugs, the feeding protocol can be modified, but a long gap between feedings is harmful to cells.Drugs are dissolved in 200 µL media per chamber, and caution must be taken while feeding the culture. The old media is aspirated out from each chamber slowly, and fresh media is added along the wall of the chamber to prevent the damage to the 3-D cultured matrix.


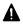
** CRITICAL STEP:** The polymerized matrix is very fragile in nature. A slight disturbance can break the matrix and spoil the 3-D culture. Pipetting in and out must be very slow and along the wall of the chamber. Pipetting from one corner of the chamber in a slightly tilted position of culture slide reduces the chances of rupture of the matrix. Multiple pipetting events in every step from the beginning of culture to downstream processing of 3-D culture product makes pipetting a very critical procedure.

Culture is fed until the desired 3-D culture morphology is achieved; 7 to 8 days are ideal for downstream procedures such as culture fixation for immunostaining (see Section 3.5 and Section 3.6) or cell extraction (see Section 3.7) for protein and RNA analyses when cells are healthy and show optimal contractions.

### 3.4. Analysis of Spheroid Cell Bodies and Fibers Formation in 3-D Cultures

#### 3.4.1. Grading of Spheroid Cell Bodies Based on Size

Spheroid cell bodies in 3-D culture environment can be graded based on their size, beating activity, and presence or absence of fiber formation. Visual inspection under the light microscope can be utilized for subjective grading of cell bodies.More accurate classification of spheroid cell bodies based on size can be done by measuring the area of these cell bodies from digital images taken in a blinded fashion. ImageJ software can be used in analysis of the area of the developing cell bodies. Download the software from the ImageJ website (http://rsb.info.nih.gov/ij/; accessed on 19 October 2021) and follow the instructions given in the user manual.Capture the images from random fields of cell culture under the light microscope. For the measurement of area, set the scale in ImageJ software prior to analysis of any image and set measurement select. Then open the image and draw an area around the cell body with the polygon selection tool and analyze the area.

#### 3.4.2. Grading of Spheroid Cell Bodies Based on Beating Activities

Visual observation under the microscope in temperature-controlled environment can differentiate the non-beating cell bodies from contracting cell bodies.Additionally, beating cell bodies can be graded based on the number of beats per minute (BPM) or percentage of beating cell bodies. This type of analysis can be used to predict the effect of a drug on ventricular cell differentiation and VCS cell development.

#### 3.4.3. Grading of Spheroid Cell Bodies by Fiber Formation Analysis

Fiber formation can be analyzed based on the number of outgrowths or fibers arising from each spheroid cell body in E11.5 ventricular cells grown in 3-D cultures.Mean fiber numbers can be determined by blinded analysis of digital images captured using a 20× phase contrast objective from 6–7 random fields/group in each experiment.Differences in fiber formation and distribution can be quantified by comparing the mean fiber numbers of spheroid cell bodies grown in 3-D cultures treated with different exogenous agonists and antagonists.

**OPTIONAL STEP:** The fiber scoring system is a subjective method to analyze the differences in the fiber formation in 3-D cultures. Scoring assignment can also be done as per the individual’s need to determine differences between groups. Other factors can also be taken in consideration to analyze any differences in fiber formation, such as fiber connections between two cell bodies, number of collateral branches arising from each outgrowth and beating activities of fibers.

### 3.5. Culture Fixation

Aspirate the media slowly from the chamber and rinse the culture with an equivalent volume of PBS at room temperature.Culture can be fixed by 4% paraformaldehyde (PFA) at room temperature.


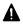
** CRITICAL STEP:** Fixation with cold 4% PFA can lead to the loss of matrix integrity, and loss of 3-D cell structures. Therefore, fixation solution should be prewarmed to the room temperature.

After 10 min of fixation at room temperature, treat the fixed cultures with PBS-glycine for 5–10 min to decrease non-specific binding during immunofluorescence staining. Wash the culture with PBS and then process for immunostaining. Fixed culture can be stored in PBS at 4 °C for up to four days.


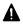
** CRITICAL STEP:** Longer storage of fixed culture in PBS is not advisable as storage for more than four days causes loss of matrix integrity, and 3-D cell structure cannot be visualized. Immediate or next day processing yields a better outcome.

### 3.6. Immunofluorescence Staining of 3-D Culture

Aspirate the PBS from the fixed 3-D culture chamber and wash the slide with PBS-glycine at room temperature for 10 min.After three washings with PBS-glycine, block the slide with 100 µL of immunofluorescence (IF) blocking solution. Cover the chambers with parafilm and incubate for 90 min at room temperature.Aspirate the IF blocking solution and add primary antibodies in 100 µL of immunofluorescence (IF) buffer. Incubate for 2 h at room temperature or overnight at 4 °C.Wash the culture 3 × 20 min with IF buffer. Add secondary antibodies diluted in 100 µL of IF buffer and incubate for 45 min at room temperature.Wash the culture with IF buffer for 20 min and then 2 × 10 min wash with PBS at room temperature.Counterstain the nuclei with Hoechst dye and incubate for 5 min.Wash the culture with PBS for 10 min and carefully remove the chamber walls without disturbing the stained culture.Spread a drop of mounting medium over the slide and put coverslip. The thickness of the matrix bed could pose some difficulty in the setting of the coverslip. A very gentle pressure without disrupting the matrix may be applied to set the coverslip.A stained slide can be stored in a dark box at −20 °C for months. A confocal microscope is ideal for imaging of 3-D cultures.

### 3.7. Cell Extraction from 3-D Culture and Protein Extraction

Aspirate the culture media from the chamber and rinse the culture with ice-cold PBS twice. Add 500 µL of ice-cold PBS-EDTA to each chamber and disrupt the matrix using the pipette tip. Gently scrape the matrix and cells from the bottom of the chamber and shake the slide for 15–30 min.Transfer the disrupted matrix solution to the conical tube. Wash the chamber with 100 µL of ice-cold PBS-EDTA and transfer it to the tube.Put the tube in the ice and gently shake for 15–30 min. Look for the complete disappearance of the BME matrix in the solution. Shake the tube till no fragments of the matrix are left, and a homogenous suspension is formed.Centrifuge the tube at 1600 rcf for 4 min in 4 °C. A small cell pellet is formed. Discard the supernatant carefully without disrupting the cell pellet. This cell pellet can be stored at−80 °C for protein extraction in the future.For protein extraction, give one additional wash to the cell pellet with PBS-EDTA to ensure the complete removal of the BME matrix.Wash the cell pellet with cell wash buffer. Centrifuge the cell suspension at 4000 rpm for 4 min in 4 °C.Aspirate the supernatant and add 50–100 µL of lysis buffer. Bigger pellet size requires a larger volume of lysis buffer. Mix the cells in lysis buffer properly and put the tube in the ice for 10 min.Sonicate the cell suspension, 3 × 10 s with 10 s break on ice. Leave the tube in the ice for 15 min after sonication.Centrifuge the suspension for 15 min at 13,000 rpm in 4 °C. Collect the supernatant into a new tube. Isolated protein can be stored at −80 °C for future use.Western blotting can be done to analyze the protein by following a standard protocol.


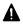
 **CRITICAL STEP:** Cell extraction from 3D cultures could be for different purposes. For the sub-culture of cells, the cell extraction procedure should be gentler. Avoid forming a tight pellet. Cells are resuspended in culture media and can be plated to the fibronectin coated dish or slides. In our experience, the yield of extracted cells from 3-D matrix for subculture is very low, but a better result can be achieved by more intense optimization of cell extraction and subculture procedures.

### 3.8. 3-D Cell Culture over the Matrix Bed in a 96-Well Culture Plate

We standardized the 3-D cell culture over the matrix bed method for a 96-well culture plate. This method gives us the feasibility for in-cell western for cells cultured in a 3-D environment.Follow the protocol for 3-D cell culture of cells over the matrix with the adjusted volume of BME matrix in each well. Coat pre-chilled 96-well culture plate with 30 µL of BME matrix. Let the coated surface dry for 15–20 min in a humid environment at 37 °C.After incubation of pre-coated wells, pipette the embryonic ventricular cells and media over the matrix. The optimum seeding concentration for each well is 50,000 cells in 150 µL of culture media. Incubate the culture plate at 37 °C and 5% CO_2_.

### 3.9. Marker Protein Expression in 2-D and 3-D Cell Culture by In-Cell Westerns

The culture surface is prepared simultaneously for both 2-D and 3-D cell culture in a 96-well plate. The 2-D culture wells are coated with 1–5 µg fibronectin/cm^2^ surface area. The 3-D culture wells are coated with BME Matrix as described earlier.A total of 50,000 cells in 150 µL of 10%FBS-DMEM are seeded in each well. Cell number and media volume is equal for both 2-D and 3-D cultures.Incubate the culture plate at 37 °C in a 5% CO_2_ environment for the desired period with regular cell culture feeding and drug of interest treatment.After the completion of the period of cell culture, follow standard protocol of immunostaining of 2-D culture in 2-D culture wells. Wash the cell culture wells with sterile PBS twice. Fix the cells in 2-D wells with 4% paraformaldehyde for 15 min at room temperature. Rinse the wells with PBS twice and permeabilize the cells with 0.1% Triton X-100 for 5 min. Further, rinse the wells with PBS twice and put 50 µL of blocking buffer (1%BSA and 10% goat serum in PBS) in each well for 1 h at room temperature.For 3-D cell culture wells, follow the culture fixation and immunostaining protocols described for 3-D cultures in Section 3.6 and Section 3.7.A separate permeabilization step with 0.1% Triton X 100 is not required for 3-D cultures, since the IF buffer used for washings, incubation of primary and secondary antibodies has 0.2% Triton X-100.Wash the culture with PBS and incubate the cell culture with primary antibodies for 2 h at room temperature or overnight at 4 °C. Wash the cell culture 3 × 15 min with PBS and add DyLight conjugated secondary antibodies for 1 h at room temperature. Wash the cell culture wells with PBS twice.Simultaneously, perform immunostaining for 3-D culture wells. Follow the protocol of fixation and immunofluorescence staining of 3-D Culture described earlier with primary and secondary DyLight conjugated antibodies.Capture the image and analyze the in-cell westerns using the LI-COR Odyssey imaging system (Channels 700 and 800).


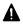
** CRITICAL STEP:** Prepare background wells for both 2-D and 3-D cell cultures with a similar number of cells. Background wells will have no primary antibodies and be added with secondary antibodies only. This helps in correction of raw values obtain by imaging system and avoid non-specific staining.


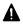
** CRITICAL STEP:** Following in-cell westerns, stain the wells with Hoechst dye and determine the cellular DNA content using a fluorescence plate reader (BMG Labtech Polarstar Omega). Cellular content in each well can be used to normalize the respective in-cell western signals.

### 3.10. Statistical Analysis

Statistical analysis was performed using Graphpad Prism version 7 (Graphpad Software, San Diego, CA, USA). All data were compared using an unpaired Student *t*-test or the ANOVA with Tukey’s multiple comparisons test and are presented as mean ± standard error of the mean (SEM). Differences of *p* < 0.05 were considered significant.

## 4. Results and Discussion

The 3-D cell culture methodologies provided an opportunity to observe the cell growth in a more physiologically relevant environment. For instance, normal 2-D cell cultures of neonatal or ES derived cardiomyocytes do not allow formation of elongated cardiac fibers under limited surface adherent conditions, whereas efficient cardiac fiber formation was reported in 3-D cell culture models [32]. In this study, we reasoned that 3-D cell culture conditions can improve embryonic ventricular cell differentiation into VCS cells. We tested three different types of BME matrices (Cultrex BME without reduced growth factors, Cultrex or Geltrex BME with reduced growth factors) and two different 3-D cell culture methods (cells on the matrix bed Vs. cells embedded in the matrix) for optimal embryonic ventricular cell differentiation studies. Cell seeding density has a major impact on the formation of spheroid cell bodies and connecting filamentous fiber formation in 3-D cultures [33]. Varying cell numbers (125,000–250,000 cells per chamber) were tested with a 70–100 µL matrix for optimal results. A lower cell number (<125,000 cells/chamber) resulted in a smaller size of spheroid cell bodies, and filamentous connections between them were rarely noticed. Cultures with a higher cell seeding densities (>250,000) led to the formation of giant cell clusters and revealed adverse effects on culture viability and beating quality over a 6–8-day culture period. Reduced nutrient diffusion to cells present in the core of the large clusters could be a possible reason for the reduced beating quality and shorter life of larger cell clusters as proposed in other studies [34]. Seeding of 125,000 to 250,000 E11.5 ventricular cells per chamber consistently resulted in the formation of spheroid cell bodies and connecting fibers with optimal viability and contractility.

Notably, the development of 3-D cell structures varied between the two cell cultures methods described here. Cells embedded in the matrix method consistently resulted in the formation of smaller spheroid cell bodies with limited inter-spheroid connections, cell contractility and viability (Appendix A) when compared to the cells on top of the matrix bed method (Figure 1, Appendix A). Reduced mobility of cells within the matrix could be one of the potential reasons for cells not able to form larger clusters using the embedding method. Reduced viability of spheroid cell bodies in the embedding method was evident after 7 to 8 days culture by the loss of contractility and black discoloration. Whereas in the case of the cells-on-top method, robust and well synchronized contractions were evident even after 13–15 days of culture. As a result, the cells-on-the-matrix-bed method was opted for all subsequent experiments.

We observed uniform distribution of E11.5 ventricular cells throughout the BME matrix using cells over the matrix bed method (Section 3.3) immediately after plating (Day 0, Figure 1A). Organized cell clusters began to form by day 2 and subsequently contractile spheroid structures developed by day 4 in all three different types of BME matrices tested (Figure 1B–D). Notably, cells grown on Cultrex BME matrix without reduced growth factors formed larger clusters at day 4 when compared to cells grown on either Cultrex or Geltrex BME matrix with reduced growth factors (Figure 1C,D). The spheroid cell bodies started to develop fibrous outgrowths by day 6 and were subsequently connected either by thin or thick filamentous contractile structures by day 8 (Figure 1E–G). We observed the formation of thick filamentous connecting structures more frequently near the chamber’s wall (Figure 1G). At this point, we do not know the possible reasons for the higher attachment of filaments to the walls of culture vessels. In addition to inter-spheroid connecting fibers, the spheroid structures also developed thin fibrous outgrowths/extensions by day 8 in culture (Figure 1H). Ventricular cells grown in BME matrices with reduced growth factors formed fewer fibrous outgrowths from spheroid structures and revealed less beating activity when compared to regular Cultrex BME (without reduced growth factors) after day 8 in culture (Appendix A; also compare Appendix A). As a result, the regular Cultrex BME matrix in combination with cells on the matrix bed method was routinely used for all subsequent experiments.

The effects of exogenous addition of ANP (1 μg/mL) and or an NPRA inhibitor A71915 (A7, 1 μM) on the growth and contractility of embryonic ventricular cells grown in 3-D cultures were determined (Figure 2A,B). The 3-D cultures were treated with these agents on alternate days for 8 days (see the timeline chart in Table 1). Previous studies demonstrated the critical role of ANP and its high affinity receptor NPRA in the embryonic ventricular cell proliferation and differentiation [11,14] as well as in cardiac hypertrophy and other disease states [35]. While ANP treatment significantly increased the mean area and the number of contractile spheroid cell bodies, co-treatment of ANP and A7 led to significant decreases in growth and contractility of spheroids cell bodies (Figure 2A,B). Furthermore, ANP treatment significantly increased the average number of fibers originating from the spheroid cell bodies with or without A7 co-treatment when compared to the control 3-D cultures (Figure 2C).

Previous molecular modeling studies showed that both A7 and ANP can interact with some common and many distinct amino acid residues in the NPRA-extracellular binding domain (ECD) dimers and subsequently stabilize the ECDs in which the juxta-membrane regions are in the proximity of each other but in opposite orientations. These differential confirmations are critical for either stimulation or inhibition of the cytoplasmic guanylyl cyclase domains [36,37,38]. Moreover, A7 was shown to act as competitive inhibitor of the radiolabelled ANP binding to the NPRA receptor [38,39]. As the ligand and antagonist were added at the same time to 3D cultures in this study, it is plausible that the net receptor conformational changes induced by A7 by replacing the bound ANP could lead to more pronounced inhibitory effects and thus cause reductions in the spheroid size and contractility when compared to ANP or A7 effects alone. While the absence of inhibitory effects on mean fiber numbers seen with combined addition of A7 and ANP (Figure 2C) may not be explained by the receptor conformational changes but could be explained by the existence of cGMP-independent NPRA signaling pathways [40]. Additional studies are required to identify molecular or biophysical mechanisms responsible for the differential effects observed with the combined addition of ANP and A7.

We next examined the suitability of 3-D culture system for VCS cell differentiation studies. Several growth factors, signaling pathways and transcription factors have been shown to play a critical role in cardiac conduction system cell differentiation [41,42,43]. The gap junction protein Cx40 expression has been used as a valid marker for VCS cell differentiation studies [11,44,45]. Confocal imaging of fixed spheroid cell bodies revealed a robust Cx40 immunostaining within the cell bodies as well as within the connecting fibers after 8 days in culture (Figure 3A–C). During cardiac morphogenesis, atrial specific gene expression (e.g., MLC-2a) was shown to be irreversibly downregulated in the developing ventricular myocardium [46]. In this study, cells were isolated from E11.5 ventricles after removing both atria and the outflow tracts. As a result, Cx40 expressing cells observed in this study represent exclusively ventricular derived conduction system cells and are unlikely to be derived from atrial cells.

A direct comparison of E11.5 ventricular cells cultured under 2-D and 3-D conditions by western blot analysis indicated a significantly higher level of Cx40 expression (2-fold) in cells cultured for 8 days under 3-D conditions compared to those cultured in 2-D conditions (Figure 3D,E). Subsequently, an in-cell western detection technique was tested for marker protein expression levels in 2-D versus the 3-D cultures grown in microwell culture plates. In-cell western analysis also revealed significant increases in the expression of Cx40 and sarcomeric myosin protein levels in cells cultured under 3-D conditions compared to cells grown in 2-D cultures for 8 days (Figure 4, normalized for cellular DNA content).

## 5. Conclusions

In this paper, we have summarized detailed protocols for efficient differentiation of embryonic ventricular cells into VCS cells using 3-D culture methods. Our results suggest that 3-D cultures can generate a larger number of VCS cells and promote higher levels of myocyte maturation compared to 2-D cultures and are better suited to validate the effects of different agonists and antagonists to study developmental pathways.

## Figures and Tables

**Figure 1 mps-05-00050-f001:**
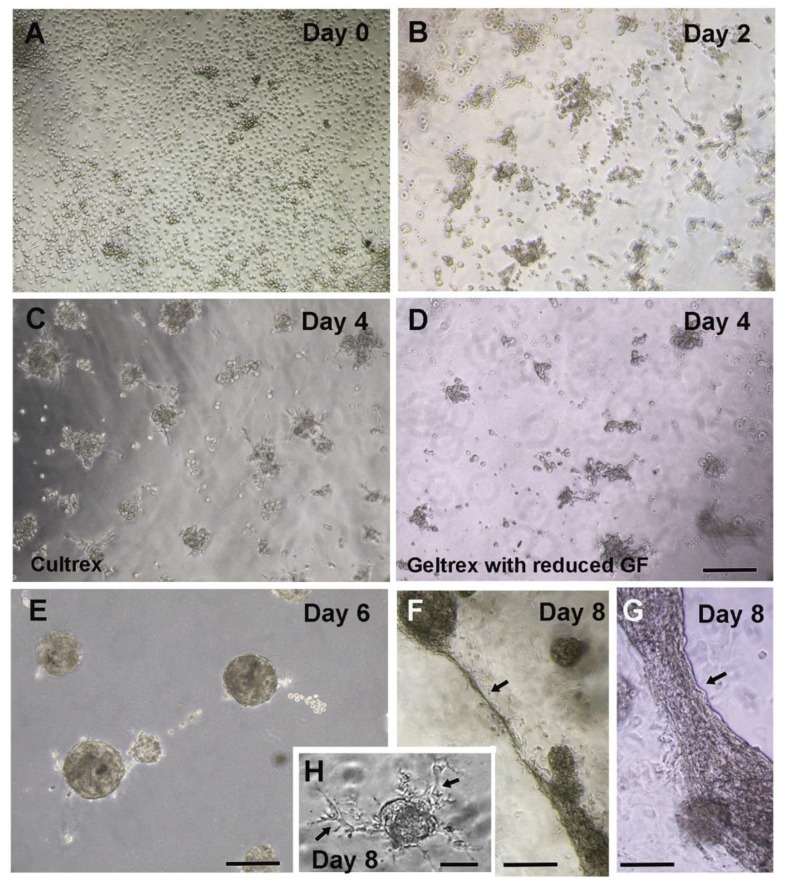
Representative 3-D culture images showing E11.5 embryonic ventricular cells in “cells over the matrix bed method” using Cultrex BME with regular growth factors (**A**–**C**,**E**–**H**) or Geltrex BME with reduced growth factors (**G**,**F**) (**D**). (**A**) At day 0, cells are uniformly spread over the matrix immediately after plating. (**B**) At the day 2 stage of culture, cells start to gather and form small clusters. (**C**,**D**) Day 4 stage of cultures show formation of larger or smaller spheroid cell bodies in Cultrex BME (**C**) and Geltrex BME with reduced GF (**D**). (**E**) At day 6, fibrous extensions begin to form between cell bodies. (**F**) At day 8, spheroid cells bodies are connected by contracting thin (**F**) or thick (**G**) filamentous structures indicated by arrows. (**H**) In addition to connecting fibers, spheroid cell bodies also develop multiple fibrous outgrowths/extensions at day 8 (indicated by arrows). Note: all images were acquired using a 10× objective; Scale bar = 100 μm for panels (**A**–**G**) and 50 μm for panel (**H**).

**Figure 2 mps-05-00050-f002:**
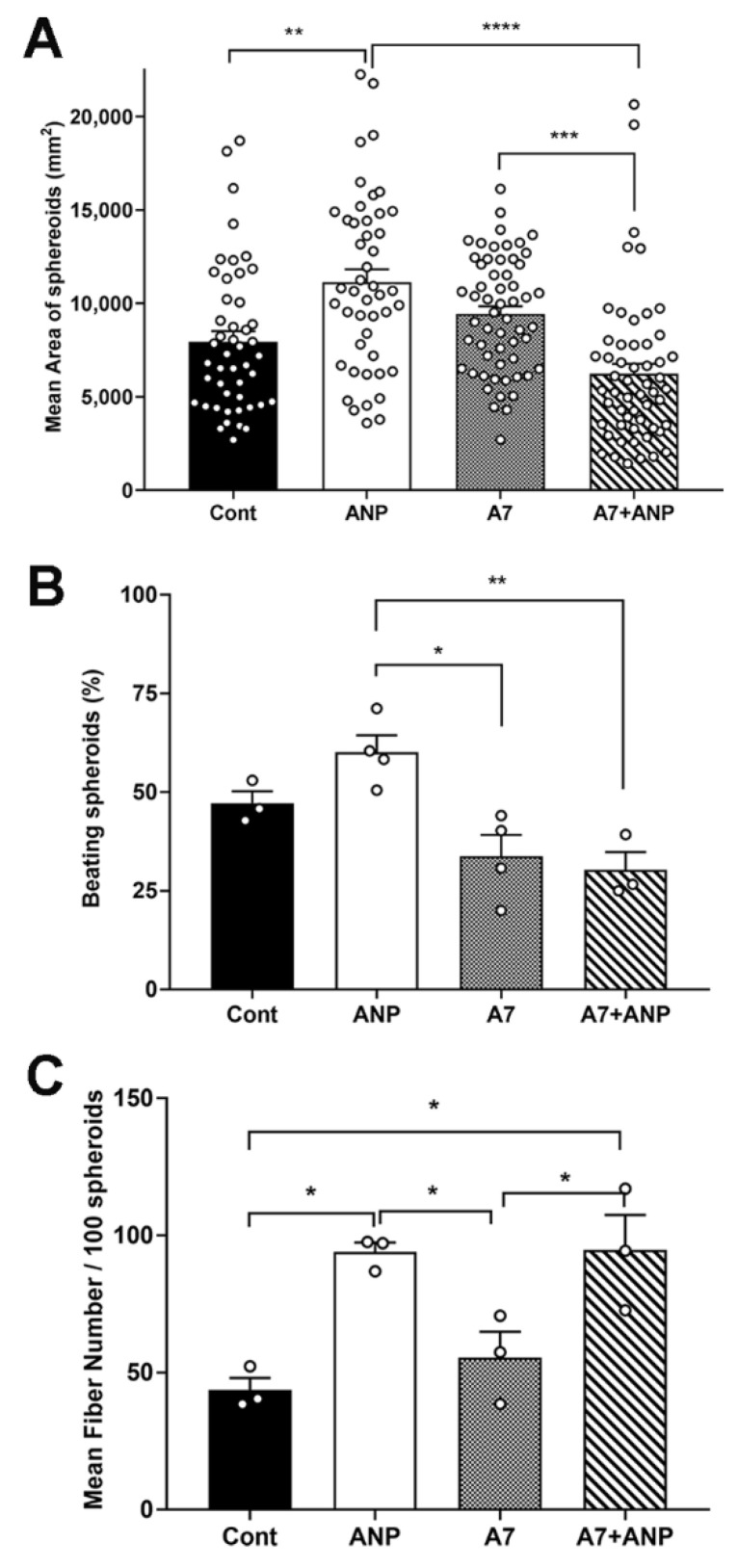
The effect of exogenous ANP (1 µg/mL) and or NPRA inhibitor, A71915 (A7; 1 μM) on 3-D cultures of E11.5 ventricular cells treated for 8 days. (**A**) Quantification of the mean areas of spheroid cell bodies, N = 3 independent experiments, 120–150 spheroid cell bodies for each group were assessed from all three experiments. (**B**) Percentage of beating spheroid cell bodies, N = 3–4 independent experiments, 9–12 random fields for each group were assessed from all three experiments. (**C**) Mean fiber numbers in 3-D cultures. N = 3 independent experiments, 100 spheroid cell bodies for each group were assessed from all three experiments. Panels A–C: Each bar represents mean ± SEM. Open circles represent individual data points for each group. * *p* < 0.05, ** *p* < 0.005,*** *p* < 0.0005 and **** *p* < 0.0001 for indicated group comparisons, One-way ANOVA with Tukey’s post hoc test.

**Figure 3 mps-05-00050-f003:**
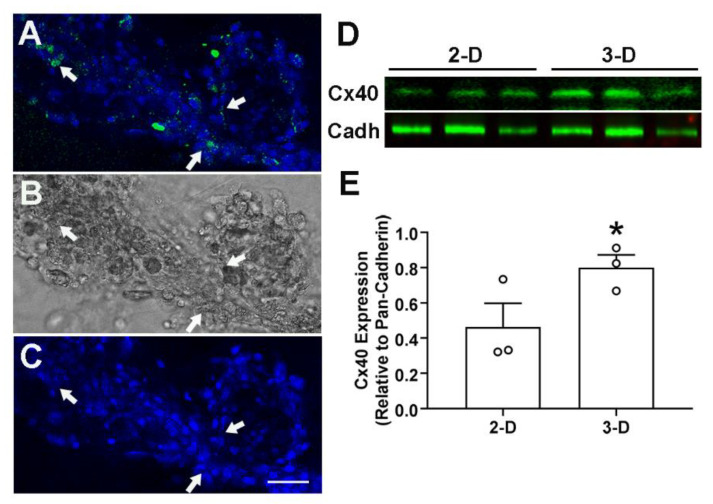
Analysis of Cx40 expression (ventricular conduction system (VCS) marker protein) in 2-D and 3-D cultures of E11.5 ventricular cells. (**A**) Representative merged confocal image of 3-D cultures stained with Cx40 antibodies (green, arrows) and Hoechst nuclear dye, (**B**,**C**) The same field was imaged under brightfield (**B**) or Hoechst fluorescence alone (**C**). A–C: Arrows indicate the relative location of Cx40 expressing cells, Bar = 50 μm, (**D**) Western blot analysis of Cx40 in membrane lysates extracted from three independent samples each from E11.5 ventricular cells cultured in 2-D (N = 3) and 3-D (N = 3) conditions for 8 days, Lower panel represents pan-cadherin (Cadh) expression levels which was used to normalize the relative levels of Cx40, (**E**) Signal intensities for samples shown in panel D were quantified using the acquisition software from Odyssey DLx (LI-COR) and Cx40 expression levels relative to the pan-cadherin (Cadh) were shown. N = 3 independent extractions for both 2-D and 3-D cultures. Each bar represents mean ± SEM. Open circles represent individual data points for each group. * *p* < 0.05 for 2-D Vs. 3-D, Student’s unpaired *t*-test.

**Figure 4 mps-05-00050-f004:**
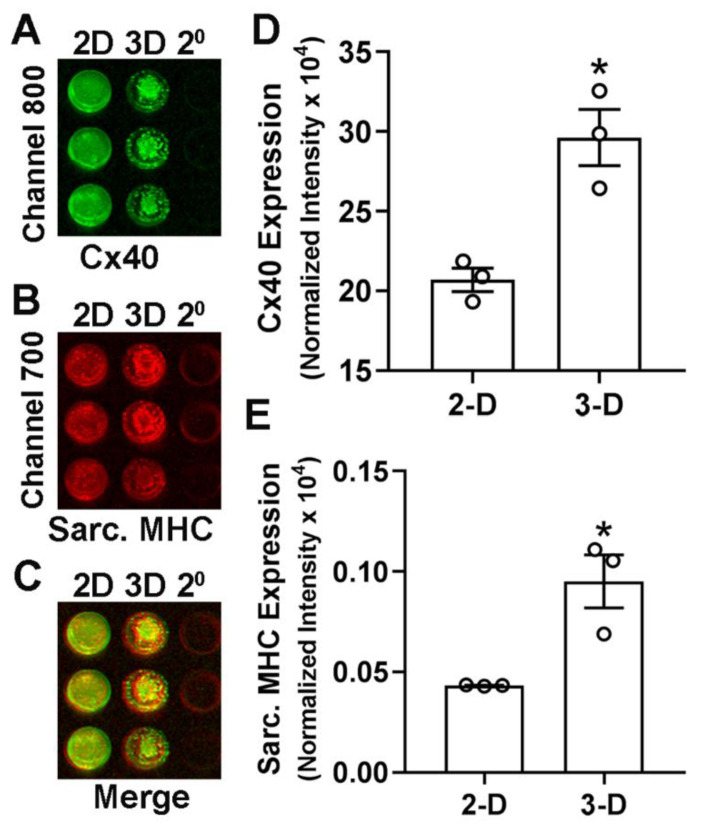
Analysis of Cx40 and sarcomeric myosin (sarc. MHC) expression levels by In-Cell western blotting technique. (**A**–**C**) E11.5 ventricular cells cultured in 2-D or 3-D conditions for 8 days were incubated with primary antibodies specific for Cx40 and sarc. MHC and signals were visualized with anti-rabbit ((**A**), 800 channel, Cx40) and anti-mouse ((**D**), 700 channel, sarc. MHC) secondary DyLight conjugated antibodies. Panel C represents merged signals from both channels shown in panels A and B. Last columns in panels A-C represent 3-D cultures stained with only secondary antibodies (2°) and omitting the primary antibodies. (**D**,**E**) Quantification of Cx40 and sarcomeric myosin heavy chain (sarc. MHC) expression levels in 2-D and 3-D cultures of E11.5 ventricular cells. Expression levels were normalized to cell number/DNA content in all wells. Each bar represents mean ± SEM. Open circles represent individual data points for each group. * *p* < 0.05 for 2-D vs. 3-D, Student’s unpaired *t*-test.

**Table 1 mps-05-00050-t001:** A timeline chart showing the details of drug concentrations and timepoints for media and drug changes. ANP: Atrial Natriuretic Peptide, A71915 or A7 (Natriuretic peptide receptor A/NPRA inhibitor).

Day−1	Day 0	Day 1	Day 3	Day 5	Day 7	Day 9
Thawing of BME matrix	Embryonic Ventricular 3-D Cell Cultures	Cell culture feeding (For all experiments)	Cell culture feeding (For all experiments)	Cell culture feeding (For all experiments)	Cell culture feeding (For all experiments)	Fixation/protein/RNA extraction
		± ANP (1 μg/mL) and A71915 or A7 (1μM) for Figure 2	± ANP (1 μg/mL) and A71915 or A7 (1μM) for Figure 2	± ANP (1 μg/mL) and A71915 or A7 (1μM) for Figure 2	± ANP (1 μg/mL) and A71915 or A7 (1μM) for Figure 2	No drug addition

## Data Availability

Data and methods used in this paper are presented in sufficient details. Any additional questions should be directed to the corresponding author.

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
