# Peer review of "Application of Three-Dimensional Culture Method in the Cardiac Conduction System Research"

_mps, 2022, doi:10.3390/mps5030050_

Round 1

Reviewer 1 Report

In the manuscript "Application of Three-Dimensional Culture Method in the Cardiac Conduction System Research" by Mishra et al., the authors has detailed the steps to generate VCS from ventricular cells isolated from mouse embryos.

The authors has well described the steps involved in culturing and differentiation procedure to generate VCS. The manuscript is well structured and explained. Here are some of the points, the authors should consider to improve it.

  • In the figure 1, panel F and G, was it acquired with the same objective lenses or it has the actual thickness difference between these two pictures. Please clarify it.

  • The methods are well details, but the details about the drug addition is bit lost. It could be better represented using a timeline chart summarizing the different steps and the drug addition time points on a chart.

  • In figure 2, the authors mention that the addition of ANP , acting via NRPA increase the spheroid size and beating compared to the control, while the addition of A7, an antagonist for NRPA decreased size and contractility. However, the combined addition of A7 and ANP caused even further reduction. This is opposite to what was expected, and also based on the authors earlier studies (PMCID: PMC5932026 ). Could you elaborate more in your discussion section.

  • Although the authors had conducted in-cell westernblot for both 2D and 3D, It is not clear why the authors has only considered to conduct normal westernblot (Figure 3B) on 3D culture and not the 2D to quantify and compare the expression levels of CX40?

Author Response

We thank the reviewer for all the critical and constructive comments. Please see the attachment for our point-by-point responses.

Reviewer 2 Report

In this paper authors describe a 3D cell culture protocol for studying cardiac cell linage differentiation in vitro. Although the method is well described and detailed, some points should be changed to improve the quality of the manuscript and consider it for its publication in the journal. As it is acknowledged by the authors, Cx40 is not only expressed by conduction system cells but also by atria cells, so I specially wonder about the specificity of the protocol, How specific for the study of conduction system? Do you specifically induce the differentiation of the cell to conduction system? Can you discriminate the percentage of cells than even expressing Cx40, belong to atria cells?

What is the difference between basement membrane extract (BME) and extracellular matrix (ECM)? Is there any specific difference between BME an a general soluble ECM like Matrigel?

In the abstract, you specify that “these methodologies can also be extended for differentiation studies using other sources of stem cells such as induced pluripotent stem cells” but, Have you demonstrate that or, on the contrary, is only a hypotehesis? This important fact must be specified and clarified in the manuscript.

In the graphs, data should be changed to scatter plots and bars instead of only bars to visualize the experimental number, the dispersion of the data and so on, at the same time. In addition, should be specified whether the errors of the data are Standard Error or Deviation. Finally, a specific section for statistical analysis should be added. Regarding this point, How reproducible re the data got by this method? This should be analyzed and specified in the manuscript.

Specific comments:

  • There are 2 sections called as 3.3 in the text, please correct
  • In the last point of the first Section 3.3. should show an example of the 3D structures formation.
  • In the second point of the section 3.6 two different fixation methods are specified. Please, provide any difference or advantages of both of them.
  • In the section 3.7, additional information about the permeabilization should be added similarly to section 3.10
  • In line 464 data not currently shown should be shown as an example of the visualization of some the problems that can appear using this method.
  • Around line 486 is specified a reduction of beating activity and authors refer to the visualization od supplemental videos. Authors must provide any quantification to support this conclusion rather than leave it in the readers’ interpretation.
  • Figure 3A: please, provide a brightfield and Hoechst staining to better visualization and data interpretation.
  • Figure 3B: please, provide the WB quantification. Also, it would improve the data if you could compare these data in 2D vs. 3D cultures, in a similar way to Figure 3F 

Author Response

(The authors gave the same response as above.)

Reviewer 3 Report

The manuscript is focused on description of new and reproducible 3-D cell culture method for studying cardiac cell lineage differentiation in vitro. Authors describe in detail the protocol of culturing of E11.5 mouse embryonic ventricular cells using a Basement membrane extract (BME) matrix to study VCS cell formation.

The paper is focused on important topic and is well written. The protocols and methods are  described clearly and contain all necessary information. Moreover, the presented images and data showing comparison of 2-D and 3-D cultures clearly document benefits of described 3-D culture methods.

I have only one minor comment - in manuscript are not complete affiliations of authors

Author Response

We thank the reviewer for all the complementary and constructive comments. Please see the attachment for our point-by-point responses.

Round 2

Reviewer 2 Report

The new clarifications resolve my previous concerns, therefore I would recommend it for publication in the journal.